# Development and Investigation of a Grasping Analysis System with Two-Axis Force Sensors at Each of the 16 Points on the Object Surface for a Hardware-Based FinRay-Type Soft Gripper

**DOI:** 10.3390/s24154896

**Published:** 2024-07-28

**Authors:** Takahide Kitamura, Kojiro Matsushita, Naoki Nakatani, Shunsei Tsuchiyama

**Affiliations:** Department of Mechanical Engineering, Gifu University, Gifu 501-1193, Japan; matsushita.kojiro.h7@f.gifu-u.ac.jp (K.M.); nakatani.naoki.i8@s.gifu-u.ac.jp (N.N.); tsuchiyama.shunsei.e6@s.gifu-u.ac.jp (S.T.)

**Keywords:** three-dimensional printing, FinRay effect, force sensors, grippers, grasp configuration, performance evaluation, soft robotics

## Abstract

The FinRay soft gripper achieves passive enveloping grasping through its functional flexible structure, adapting to the contact configuration of the object to be grasped. However, variations in beam position and thickness lead to different behaviors, making it important to research the relationship between structure and force. Conventional research using FEM simulations has tested various virtual FinRay models but replicating phenomena such as buckling and slipping has been challenging. While hardware-based methods that involve installing sensors on the gripper and the object to analyze their states have been attempted, no studies have focused on the tangential contact force related to slipping. Therefore, we developed a 16-way object contact force measurement device incorporating two-axis force sensors into each of the 16 segmented objects and compared the normal and tangential components of the enveloping grasping force of the FinRay soft gripper under two types of contact friction conditions. In the first experiment, the proposed device was compared with a device containing a six-axis force sensor in one segmented object, confirming that the proposed device has no issues with measurement performance. In the second experiment, comparisons of the proposed device were made under various conditions: two contact friction states, three object contact positions, and two object motion states. The results demonstrated that the proposed device could decompose and analyze the grasping force into its normal and tangential components for each segmented object. Moreover, low friction conditions result in a wide contact area with lower tangential frictional force and a uniform normal pushing force, achieving effective enveloping grasping.

## 1. Introduction

Soft grippers have been attracting attention in recent years, and various types of soft grippers have been developed. The structure of a soft gripper is composed of flexible materials, and the gripper itself can deform according to the shape of the object to be grasped. A typical example of a soft gripper is the FinRay soft gripper based on FinRayEffect^®^, which was developed in 1997 by Leif Kniese et al. in their study of fish fins. FinRayEffect^®^ is a geometric structure consisting of a triangular front beam, a back beam, and a crossbeam connecting the two (Figure 1) [1]. This geometric structure is called the FinRay structure, and the contact force from outside the front beam deforms the entire structure, resulting in an action that envelops the contacted object. We refer to the grasping motion utilizing this phenomenon as “enveloping grasp”, characterized by the distribution of contact forces over a wide area to achieve stable object grasping. To properly perform enveloping grasp, it is necessary to quantitatively evaluate the relationship between the normal force applied in the normal direction to the contact surface, the frictional force applied in the tangential direction, and the amount of structural deformation. The FinRay structure has been released as open source [2,3] and is easy to implement in robotic systems, such as object manipulation with simple control and modeling with a 3D printer. Therefore, a wide variety of applications have been proposed; for example, some research has introduced the FinRay structure into systems for grasping agricultural crops and various daily necessities [4,5]. These studies have focused on the ability to stably grasp various objects with one type of FinRay structure geometry. However, the FinRay soft gripper slips and wraps around the contact surface of the object to be grasped, and no studies have quantitatively evaluated or discussed the changes in grasping properties due to “slipping” or “buckling” of the FinRay structure. In order to analyze these phenomena, it is important to evaluate the distribution of contact forces, such as pinch force and friction force, applied to the object to be grasped and structural deformation characteristics.

Finite element method (FEM) simulation is a prevalent evaluation method for FinRay soft grippers and has been used in many studies [6,7,8,9,10,11]. However, as Jack et al. [12] showed, the accurate verification of “buckling” and “slipping” is difficult with FEM simulations and realizing a methodology that shows the same results in both simulation and reality is a challenge.

Therefore, this study aims to propose a system capable of grasp force analysis and the measurement of grasp force and friction direction using a FinRay soft gripper, and to analyze enveloping grasp using this system. First, to confirm that this measurement system can quantitatively evaluate the distribution of normal force and frictional force, we compare its results with those obtained using a conventional six-axis force/torque sensor, commonly used in previous studies. Next, using the proposed system with confirmed measurement performance, we analyze the impact of different object grasping conditions, such as initial contact position and contact surface friction characteristics, and discuss the conditions that enable stable enveloping grasp. We believe that analyzing the relationship between object grasping conditions and enveloping grasp will contribute to the future design guidelines of FinRay soft grippers and the control strategies of grasping motions.

## 2. Related Work

Several methods for analyzing the enveloping grasp state using a FinRay soft gripper prototype, rather than simulations, have been proposed in previous studies. We classify these evaluation methods into three types based on the type and placement of the sensors used.

The first method is to install sensors directly on each beam of the FinRay structure [13,14,15]. Hongyu et al. [16] installed six piezoelectric elements on the front and back beams and on the crossbeam of a FinRay soft gripper and analyzed the internal forces at each position. Another study placed a transparent gel with known mechanical properties and assigned feature points inside a FinRay structure, and the gel deformation and feature point displacement caused by the deformation of the structure were measured using camera images [17,18,19,20]. However, there are concerns that incorporating such a sensing system would interfere with the original deformation of the FinRay structure and complicate the system.

In response to these concerns, a second evaluation method was developed, in which the force sensor was installed, not inside the FinRay structure, but inside the object to be grasped. Xiaowei et al. [21] split the object to be grasped (cylinder) into two parts and installed a six-axis force sensor at the central connection of the two parts to realize six-axis force (Fx, Fy, Fz, Mx, My, Mz) analysis, enabling pure contact force analysis of the FinRay structure. However, since this method analyzes the resultant force applied to the center point of the grasped object, it does not provide quantitative data on the state of contact force distribution due to the wrapping characteristics of the FinRay soft gripper.

The third evaluation method is to record camera images of the entire FinRay structure and analyze them to evaluate changes in the shape of the structure and estimate internal forces from its material properties [21,22,23,24]. However, since the material properties of the FinRay structure must be known in detail, estimation of its internal forces requires a more complex model than that required for a simple structure, such as a beam, and guaranteeing an agreement between the actual machine and the model is challenging.

As mentioned before, research on FinRay soft grippers mostly focuses on evaluating contact force distribution and frictional forces on contact surfaces, and not enough research has been conducted by including factors such as “buckling” and “slipping”, which are problems inherent in actual machines.

In order to improve the spatial resolution for measuring contact force distribution, in our previous study, we constructed a 16-way object contact force measurement device with 16 single-axis force sensors arranged in a circle. Furthermore, we developed a structural deformation measurement device based on camera image analysis to record the deformation of the FinRay structure. Using these two devices, grasping experiments were conducted on a parallel open/close soft gripper equipped with the FinRay structure to measure and analyze the contact conditions when grasping cylindrical and square objects [25]. As a result, we could analyze the correspondence between the buckling and grasping characteristics of each beam. However, this is not sufficient for quantitative evaluation of the relationship between frictional force and grasping characteristics, due to “slipping,” because the measurement axis direction of each force sensor of the 16-way object contact force measurement device can only analyze the center direction component of the grasped object. In other words, additional measurement axes are required to measure the tangential frictional forces at 16 points along the circumference where the force sensors are located.

Based on the above, the proposed contact force distribution measurement device with built-in force sensor object is a 32-axis measure capable of measuring normal force and friction force in two-axis direction at each 16 points and attempts to evaluate the distribution of normal force and friction force.

## 3. Materials and Methods

This section describes the 16-way object contact force measurement device we proposed and the additional axial directions that can be measured, as well as how the proposed measurement device compares to conventional ones.

### 3.1. Additional Measurable Axial Directions

As shown in Figure 2a, the object (cylinder) was divided into 16 arcs (every 22.5 degrees), and a single-axis force sensor (load cell rated capacity: 2 kg) was placed at each arc. The single-axis force sensor was set so that the measurement axis passed through the center of the cylinder. This sensor arrangement allowed the magnitude and distribution of the normal force perpendicular to the 16-segmented surface of the cylinder to be measured all at once. The cylinder consisted of a component cut from the aluminum plate shown in Figure 2b. In total, 16 contact blades were arranged in each direction of division. By replacing these blades, the diameter of the cylinder could be changed, and measurements could be made, assuming shapes other than a cylinder, such as a rectangular object. The soft gripper and contact blades were composed of Elastic 50A (RS-F2-ELCL-02, Formlabs Inc., Sommerville, MA, USA) and an aluminum plate (A5052), respectively, and friction tests determined the coefficients of friction between the contact blades and the contact surface of the soft gripper to be approximately 0.88 for static friction and 0.79 for dynamic friction.

We considered it necessary to add a measurable sensor axis direction to our device for frictional force measurement. Since our previous studies have shown that the normal force on the blade contact surface tends to be about 5 N [25] and the associated static friction force is about 4.4 N, a small single-axis force sensor (load cell rated capacity: 500 g) was newly incorporated into the device in the measurement section of the contact blade and force sensor, as shown in Figure 2c. Using PLA resin with an FDM 3D printer, we newly fabricated a jig to fix the small force sensor so that its measurement direction was perpendicular to the direction of the normal force and a jig to fix the blade (Figure 2b, right).

As shown in Figure 3, the bottom of the contact force measurement device was fixed to the aluminum frame with a dedicated aluminum plate. The method of fixing the device between the aluminum plate dedicated for fixing and the aluminum frame of the device base could be switched between simple screw fixation (fixed) and fixation using a slider mechanism (non-fixed). When the slider mechanism was used, the displacement of the cylindrical object could be measured with an accuracy of 200 μm using a micro-laser range finder (HG-C1100, Panasonic Inc., Osaka, Japan). The laser beam of the laser range finder was irradiated onto Kent paper attached to the side of the aluminum plate dedicated for fixing.

The force sensor and laser range finder readings were output as analog voltage signals, which were measured and recorded using PC 1 via Data Acquisition (DAQ: USB-6218, National Instruments Co., Austin, TX, USA). We performed 5 measurement trials for each experimental condition. Data acquisition was recorded at a sampling rate of 1 kHz, with a resolution of 16 bits, and data from the 15 trials were processed as follows: (1) read the time series data, (2) remove high-frequency noise with a Savitzky–Golay filter, (3) calculate the time of moving averages of the 15 trials in MATLAB (R2023a), and (4) display the averages in a graph.

The contact initiation position is also an important experimental parameter in the grasping motion of the FinRay soft gripper to grasp a cylindrical object. Therefore, three different contact initiation positions were defined: the center of the FinRay structure’s contact surface, +10 mm from the center of the contact surface to the tip of the FinRay structure, and +20 mm from the center of the contact surface to the tip of the FinRay structure (Figure 3c). These settings were made in both fixed and non-fixed device fixation methods.

### 3.2. Design of the FinRay Soft Gripper

The appearance and size of the FinRay soft gripper are shown in Figure 4a. The FinRay soft gripper consisted of a gripper-opening/gripper-closing mechanism and a FinRay structure. The opening/closing mechanism uses a parallel opening/closing system with rack and pinion gears. This mechanism allows for relatively easy adjustment of the opening width and gripping force using servo motors (Dynamixel XM540; Robotis, Seoul, Republic of Korea). Two motors were used in the drive system. The opening and closing mechanisms were designed in 3D using Autocad’s Fusion360 and fabricated using a UP300 FDM 3D printer with ABS resin (Tiertime Corporation, Beijing, China). The FinRay structure used for the soft gripper is shown in Figure 4b,c. The structure was based on the DHAS-GF-120 CAD data [2,3] published by Festo, Inc. (Esslingen am Neckar, Germany), and was fabricated using a Formlabs optical 3D printer Form3 (Sommerville, MA, USA). A rubber-like resin material (Elastic50A) was used for modeling. The bases for the color markers were placed on the sides of the FinRay structure during the 3D design process and were sculpted as one piece. This allowed for uniform marker positioning, even when different FinRay structures were used. The FinRay structure formed could be fixed via a special fixing jig constructed using a 3D printer. The jig was inserted into a T-slot in the movable part of the opening/closing mechanism, making it easy to attach, detach, and replace.

### 3.3. Deformation Position Measurement System Using Image Analysis

Since the experiment also evaluated the amount of deformation of the FinRay structure, a deformation position measurement system based on image analysis was used, as shown in Figure 5. The system consisted of a USB camera (ELP 4K; Ailipu Technology Co., Ltd., Shenzhen, China), an angle control motor (Dynamixel XM540; Robotis), a group of color markers, and a computer (PC2) for measuring and analyzing the camera images. As shown in Figure 4 and Figure 5, we attached 23 red markers to the nodes at the intersection of the main beam and crossbeam on the side of 1 finger of the FinRay soft gripper. Furthermore, the USB camera itself could be moved with motor control so that the bottom of 1 finger of the FinRay soft gripper was always in a fixed position in the image. A video of grasping (1280 × 720 px, 60 fps) was recorded on PC2 for measurement and analysis (Appendix A).

### 3.4. Conventional Measurement Methods and Equipment

To compare with the proposed measurement device, a measurement device using a single 6-axis force sensor was also developed. Figure 6 shows a device in which one 6-axis force sensor is placed in the center of a cylinder, the cylinder is divided into 16 sections, and triaxial forces on the surface of one of the divided sections is measured. This experiment can also be performed by rotating the cylinder’s measurement section by 22.5 degrees (360 degrees/16 divisions). By rotating the cylinder’s measurement section by 22.5 degrees for each trial of the experiment, we evaluated 16 directional contact forces when the cylindrical object was grasped. In other words, this device can be used to verify the grasping force decomposition measurement performance of 16/16-type devices. In each direction, as with the 16/16-type device, five measurements were taken and the values of the force-time graphs at each time point average were calculated. When using this method, the number of trials required is more than that in the 16/16-type device. As with the 16/16-type device, measurement data were recorded as analog signals from a 6-axis force sensor using DAQ. We analyzed the relationship between the normal force, frictional force, and structural deformation properties of the FinRay soft gripper when assuming cylindrical-object-grasping properties using these devices.

## 4. Results

### 4.1. Comparative Verification of Two Types of Built-In and Contact Force Measurement Devices for Objects to Be Grasped

Our proposed device was improved from the previous 16/16-type device, which only measured forces perpendicular to the contact surface, to also measure tangential forces. Therefore, to verify the validity of our proposed device, we compared it with a 1/16-type device using a single six-axis force sensor. The comparison focused on three aspects. The first aspect was the magnitude of normal and frictional forces under the same conditions. Since the proposed device included measurements that accounted for slipping due to friction, there were slight variations in the force magnitude under identical conditions. Hence, if the difference was within ±1 N, it was considered that similar grasping tendencies were achieved. A settling time of 5.25 s, when the signal rose due to stress concentration at the contact surface and converged within ±5%, was used as a reference. The second aspect was the range of contact force distribution in both normal and tangential directions. A wider range indicated a broader measurement range and higher spatial resolution. The third aspect was the experimental operation time required for measurement when the grasping trials were repeated five times., as the measurement operations for normal and frictional forces differ between the 1/16-type and 16/16-type devices. For comparison, experiments were conducted under four conditions, as shown in Table 1, for each device. The high-/low friction states on the contact surface were part of the experimental environment, with lubrication (Kure55-6; KURE Engineering Ltd., Tokyo, Japan) applied for low friction but not for high friction. The results are expected to vary significantly under different lubrication states. To achieve a low friction condition, a spray can was used to apply lubricant to the contact surface of the FinRay soft gripper. This method allows for uniform application. Additionally, since the lubricant film applied to the contact surface by the measurement device’s blade is wiped away after each gripping attempt, lubrication is performed before each subsequent gripping operation. In this experiment, considering that the cylindrical object was the grasping target and the gripper operated symmetrically, we analyzed the data from channels 1 to 8 on the right side of the device viewed from above. We conducted 15 measurement trials for each condition and plotted the average values of these trials to compare the differences between each condition. The results of the 1/16-type device are shown in Figure 7, and the results of the 16/16-type device are shown in Figure 8.

Figure 7 shows the distribution and magnitude of the normal and frictional forces for cylindrical object grasping with the FinRay soft gripper. Furthermore, as shown in Figure 7a,b, the maximum normal force in the high friction and low friction states was around 2 N in the channel 6 direction and the frictional force decreased from about 5 N to around 2 N, respectively. The decrease in load can be attributed to the change in friction characteristics between the contact surface of the front beam of the FinRay structure and the contact blade due to the application of lubricant, making it more slippery. Since the FinRay structure undergoes elastic deformation, a larger deformation increases the elastic energy stored within the structure. However, the slippage phenomenon releases some of this elastic energy, which leads to a reduction in localized loads that are transferred to the normal and friction forces. In addition, This slippage caused a change in the contact position under high and low friction conditions, making it difficult for the tip of the FinRay structure to contact areas farther from the gripper base. Consequently, the normal force in the channel 2 direction, following the channel 1 direction, no longer reacted, and a tendency for the contact area to decrease was observed.

As shown in Figure 8, the same tendencies observed in the 1/16-type device could be read. When comparing Figure 7 and Figure 8, in the high friction state, the magnitude of normal forces measured with both devices, except for the channel 6 direction, was within ±1 N, indicating that the 16/16-type device can measure forces of a magnitude similar to that of the 1/16-type device. The difference in the channel 6 direction was likely due to the roughness caused by processing errors on the contact surface of the FinRay structure’s blade. The distribution range of normal and frictional forces was four directions in the high friction state and five directions in the low friction state for the 1/16-type device compared to five directions in both friction states for the 16/16-type device. Thus, the measurement range and trends for both devices were similar, demonstrating equivalent performance in terms of the distribution range.

Finally, regarding the time required for measurement operations in the two devices, the 16/16-type device took 20 s for data recording for eight directions, while the 1/16-type device took the same 20 s for each segment direction but an additional 25 seconds for each axis adjustment, totaling 335 s. This resulted in a reduction of 315 s in the experimental measurement time. Therefore, the 16/16-type device is useful as it provides equivalent measurement performance to the 1/16-type device but allows for experiments to be conducted in a shorter time. In subsequent experiments, the 16/16-type device was used to analyze the relationship between normal and frictional forces and structural deformation.

### 4.2. Analysis of Frictional Force and Structural Deformation Characteristics, Depending on Gripping Position and Contact Surface Conditions

Using the 16/16-type device, we analyzed the relationship between the normal force, the frictional force, and structural deformation characteristics, depending on the contact initiation position and contact surface condition. The 16/16-type device was in a fixed position, and three different contact initiation positions and two different contact surface conditions, namely high friction (static friction coefficient: 0.88) and low friction (static friction coefficient: 0.64), were tested. In this validation, we conducted 15 trials for each condition and plotted the average values of these trials to compare the differences between each condition in Table 2.

Figure 9 shows the experimental results of settings 5–7. The peak values of the normal force and friction force distributions moved to the vicinity of the channel 5 direction as the contact initiation position approached the tip, and the magnitude of the distribution remained almost the same. When comparing the normal force and friction force vectors, we found that the friction force vector was longer than the normal force vector, indicating that the friction force is larger and that loads other than the static frictional force are being applied. In addition, the distribution of both normal and frictional forces was localized in the channel 5 and 6 directions.

Figure 10 shows the experimental results of settings 8–10. The normal force distribution in each channel direction showed a similar trend as that in the results of settings 5–7, with the magnitude of the normal force decreasing only by about 1 N. However, the frictional force decreased by a maximum of 4 N, and the vector graph showed that the vector length of the friction force was shorter than that of the normal force. This was close to the value obtained by multiplying the normal force in each channel direction by the coefficient of static friction, indicating that the friction direction components other than the static friction force may have been removed by the lubricant. Here, the structural deformation shown in Figure 10d indicates that the end position of the marker on the front beam was also the same, although the position of the cylindrical object was almost the same in the high friction (red) and low friction (blue) states because it was fixed. The fact that the movement trajectory of the marker was different near the tip suggests that low friction causes the FinRay structure to grasp in a shape that is less likely to accumulate internal stress.

From these results, we can conclude that when an object is fixed for measurement, the conventional measurement method does not allow detailed evaluation of loads other than the frictional force, and the method is effective in load evaluation by changing the contact surface friction state. However, assuming actual object grasping, stable object grasping may not be realized in a low friction state, and it is necessary to design a gripper that can adjust between low and high friction. Therefore, we believe that excessive load generation can be reduced by conducting measurements in an experimental setting that considers non-fixed, back-and-forth movement of the gripped object, rather than fixed. In addition, the environment in which the soft gripper is implemented mainly assumes the pick-and-place of the object, and often, the object is not fixed to the operating environment. Therefore, the non-fixed condition enables measurement that is in line with the actual operating environment.

### 4.3. Contact Force Evaluation Considering Object Movement

Here, we explain the grasping experiments, considering the displacement of the object’s position. The device was set to the non-fixed condition, and settings 11–16 in Table 3 were the measurement conditions for verifying differences between high- and low friction states. As in previous experiments, in this validation, we conducted 15 trials for each condition and plotted the average values of these trials to compare the differences between each condition.

First, Figure 11 shows the results for settings 11–13. The contact start positions for the grasped object correspond to L = +20 mm, L = +10 mm, and L = 0 mm in Figure 11a, b, and c, respectively. Additionally, each figure shows, from left to right, the distribution plots of the vector representations of the normal, frictional, and biaxial forces, and compares the contact force states. When the central contact surface was the initial contact point, both normal and friction forces were larger compared to those when grasping near the tip. Grasping at the midpoint may induce other loads, in addition to friction forces, as explained in settings 5–7. In contrast, when contact started at L = +10 mm or L = +20 mm, the normal-to-friction-force ratio was around 1.0 to 0.8, close to the static friction coefficient. The vector graphs of normal and friction forces showed that contact at L = +10 mm or L = +20 mm resulted in weaker forces distributed without causing localized contact forces.

Next, Figure 12 shows the results for settings 14–16. Figure 12 has a similar graph layout to Figure 11, with Figure 12d depicting the movement trajectory of the red markers placed on the front beam of the FinRay soft gripper. According to these graphs, both normal and friction forces were distributed over a wider area, with relatively smaller values, compared to those in settings 11–13. This indicates that a less loaded grasping state was achieved due to the low friction state and the object being able to move. The distribution of contact forces that cannot be fully relieved by slippage alone is likely adjusted by moving the object itself to a position where the contact forces are further reduced. We consider this to be the object manipulation function of the FinRay soft gripper. When incorporating this into a robotic system, the manipulation distance of the object should be taken into consideration. The widest contact range occurred at low friction, with an initial contact position of L = 0 mm, measuring the normal force at six locations (channels 2–7), as shown in Figure 12c. This condition maximally exhibited the FinRay structure’s enveloping grasp. However, the laser displacement in Figure 13 shows variability in movement, suggesting more slipping in the low friction state, with reduced variability at L = +10 mm and L = +20 mm. This variability poses a significant challenge in planning robot manipulation tasks and incorporating a mechanism to alter friction states within the FinRay structure is a potential solution.

## 5. Discussion

We verified three aspects using the proposed measurement system. First, we compared the measurement performance of the 1/16-type measurement device with one six-axis force sensor and the 16/16-type measurement device with 32 one-axis force sensors. The comparison focused on the spatial resolution of normal and friction forces and the time required for experimental operations. The results showed that both devices could evaluate the distribution of forces, and the experimental operation time was reduced by approximately 94% with the 16/16-type device, confirming the utility of the proposed 16/16-type system. Furthermore, when changing the friction characteristics of the contact surface between low- and high friction conditions, reducing friction contributed to the reduction of local contact forces, which is considered important for flexible enveloping grasping. However, reducing friction with simple lubricants might make stable grasping of objects difficult, indicating that switching friction characteristics according to the operation process is a future challenge for adopting this method in robotic object handling systems.

Secondly, we examined the changes in contact force distribution when varying the initial contact positions (L = 0 mm, L = +10 mm, L = +20 mm). The results showed that the closer the initial contact position was to L = +20 mm, the more it contributed to a reduction of approximately 40% in local contact forces. This suggests that changing the friction characteristics of the contact surface leads to a further reduction in friction forces. Therefore, grasping fragile objects effectively involves grasping as close to the tip of the FinRay structure as possible and reducing the contact surface friction. However, the experimental results also showed that the width of the contact force distribution decreased as the initial contact position approached L = +20 mm, making enveloping grasping impossible. This indicates the need to develop a control system that determines the contact position according to the shape of the object being grasped.

The FinRay soft gripper is primarily used for object picking tasks, and it is considered that the relative position with the target changes when grasping objects that are not fixed in the working environment. Therefore, as the third verification, we evaluated the contact force distribution under non-fixed conditions of the grasped object. The results showed that in the high friction state, except for L = 0 mm, the contact force decreased compared to the distribution results under fixed conditions, confirming that moving the object makes it difficult to accumulate elastic energy in the FinRay structure. Next, the distribution under low friction conditions showed almost the same level of contact force distribution as the results under fixed conditions, with relatively little object movement. This indicates that the impact of friction conditions is effective in reducing local contact forces, and the subsequent reduction in load due to object movement is also influential. Additionally, the object’s stopping position after grasping often deviated from the initial position, with the case of contacting at L = 0 mm under low friction conditions being the closest to returning to the initial position. Therefore, when planning object grasping and transport, it is necessary to consider this positional error in control, but there is variability in the movement distance, presenting a challenge in the predictive uncertainty of the slipping phenomenon.

## 6. Conclusions

In this paper, we introduced a force sensor-integrated contact force measurement system to evaluate the enveloping grasping characteristics of the FinRay soft gripper. The main objectives were to propose an experimental apparatus and evaluate the enveloping grasp of the FinRay soft gripper.

The featured aspect of the introduced force sensor-integrated contact force measurement system is its capability to measure 32-axis forces by adding 16 single-axis force sensors for friction measurement to the conventional 16/16-type measurement device. From the initial verification, it was confirmed that this system provides higher spatial resolution and reduced experimental operation time compared to the 1/16-type device with a single six-axis force sensor. This improvement enables more detailed contact force evaluation and enhances experimental efficiency. Subsequently, using this device, we evaluated the effects of contact surface friction characteristics, initial contact position, and the fixation state of the object on enveloping grasping. The insights gained from this evaluation provide guidelines for integrating the FinRay soft gripper into robotic systems and can be applied to the optimal design of the FinRay structure considering slippage phenomena.

## Figures and Tables

**Figure 1 sensors-24-04896-f001:**
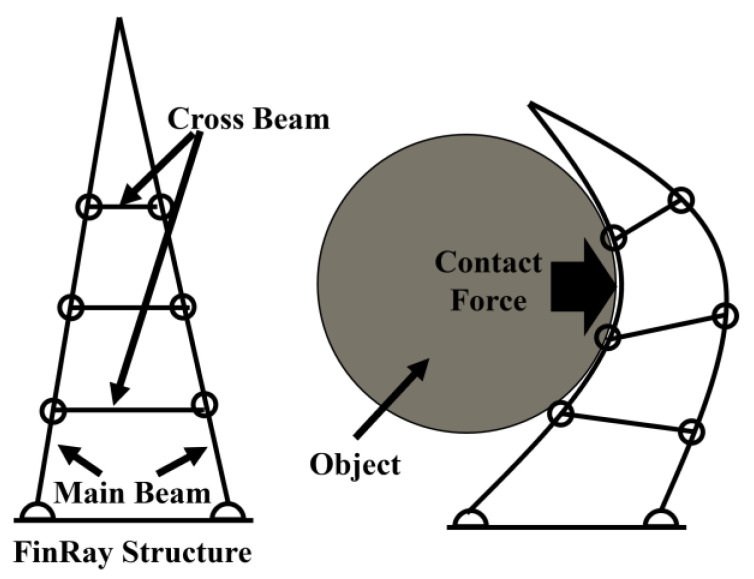
Arrangement of each beam in the FinRay structure (**left**) and the passive deformation mechanism (**right**).

**Figure 2 sensors-24-04896-f002:**
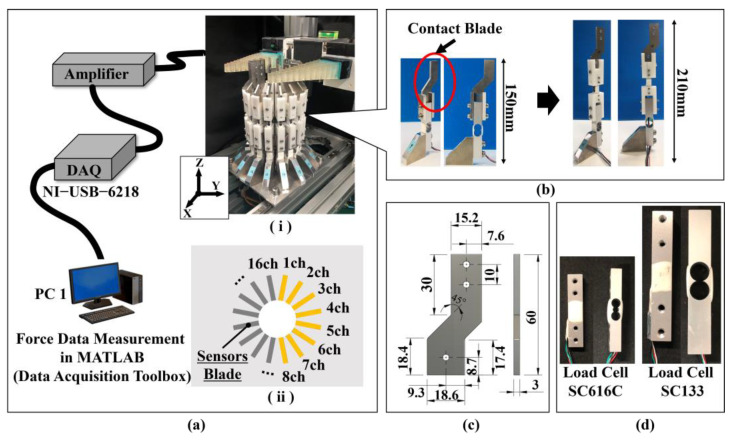
Conceptual diagram of the distributed contact force measurement device with a built-in object to be grasped. (**a**) Introduction of measurement equipment: (**i**) Overall view; (**ii**) 16 measuring point numbers, (**b**) additions and design changes in the direction of the measurement axis, (**c**) an aluminum contact blade (shape of the contact area), and (**d**) force sensor size and mounting method.

**Figure 3 sensors-24-04896-f003:**
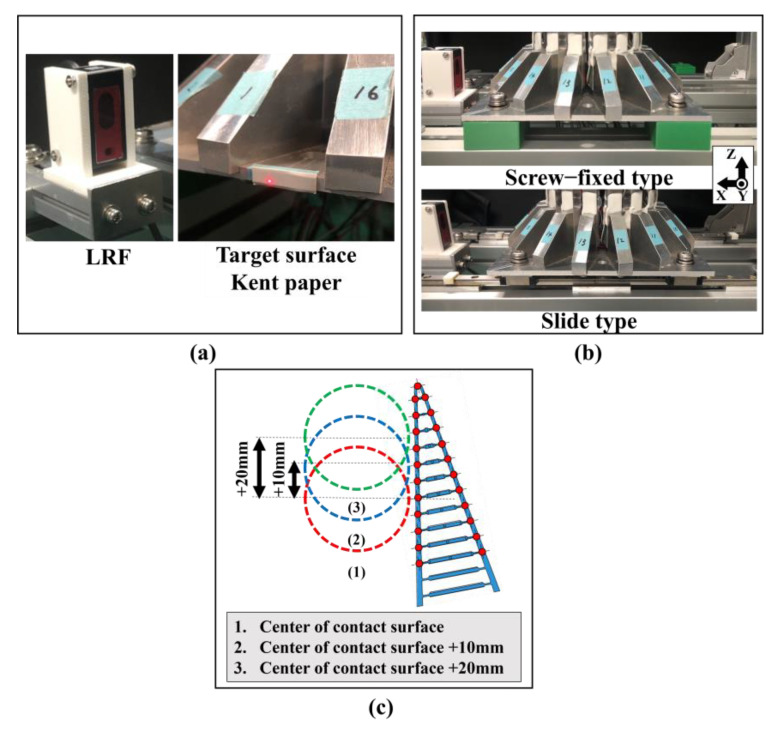
Back and forth action mechanisms for the contact force measurement device. (**a**) Installation of the laser range finder and view of the laser-irradiated surface, (**b**) switching mechanism for forward/backward movement of the object (upper image: sliding mechanism; lower image: screw-fixing mechanism), and (**c**) contact positions of three types of cylindrical objects.

**Figure 4 sensors-24-04896-f004:**
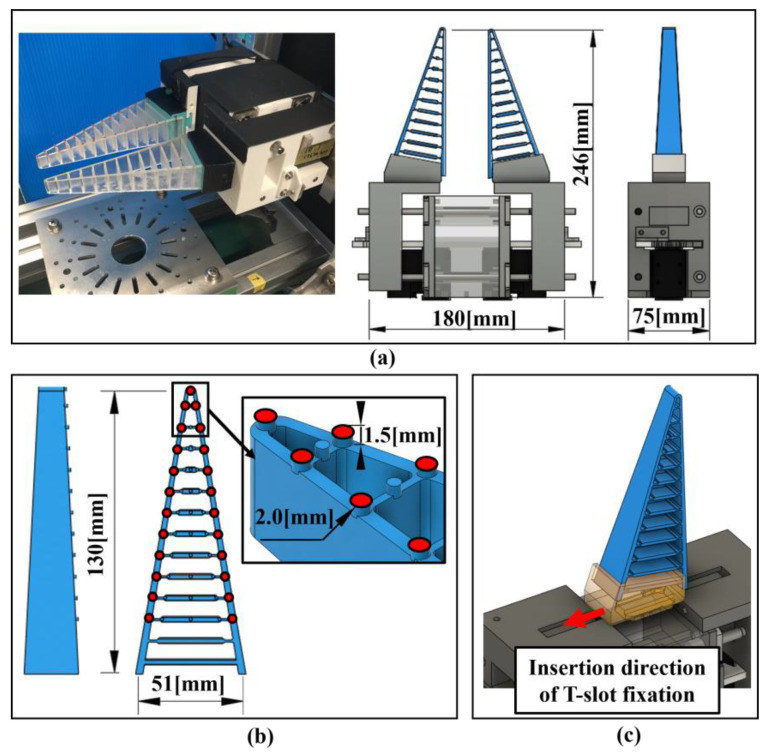
Parallel opening/closing mechanism of the FinRay soft gripper and finger design with a marker base. (**a**) Appearance of the FinRay soft gripper and the size and structure of the opening and closing mechanisms, (**b**) size of the FinRay structure and the location and size of the marker base, and (**c**) method of fixing the FinRay structure to the moving part of the opening/closing mechanism (T-slot type).

**Figure 5 sensors-24-04896-f005:**
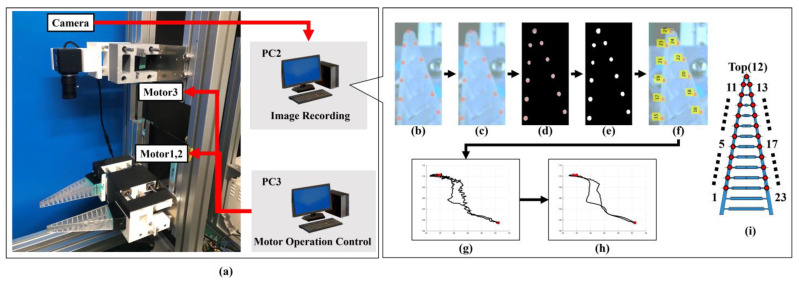
Conceptual diagram of the structural deformation measurement device using camera images and flow of image processing and analysis of marker trajectories. (**a**) Relationship between the USB camera, motor, and PC (PC3 for controlling gripper motors 1, 2, and camera base motor 3) that made up the device; (**b**) initial image; (**c**) Gaussian smoothing image; (**d**) masking image; (**e**) binarized image; (**f**) recording of the center of gravity using a Kalman filter and numbering of recognition blobs; (**g**) unit conversion of blob center-of-gravity coordinates (pixel to mm); (**h**) smoothing with Savitzky–Golay filter; and (**i**) numbering position.

**Figure 6 sensors-24-04896-f006:**
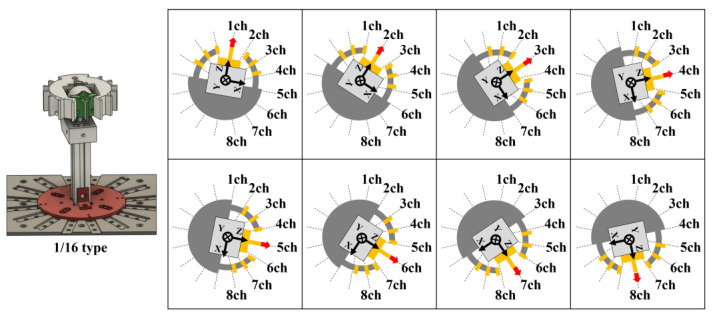
The 1/16-type measurement device using a 6-axis force sensor and the measurement method for channels 1–8 (1ch to 8ch) directions of 16 divisions (same for channels 9–16).

**Figure 7 sensors-24-04896-f007:**
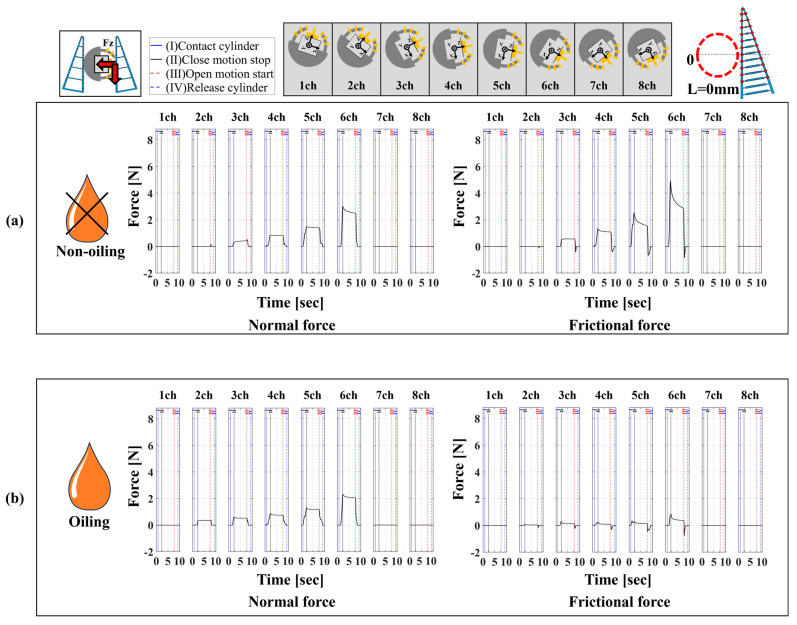
The 1/16-type measurement device using a 6-axis force sensor and the measurement method from 1ch to 8ch directions of 16 divisions (same from 9ch to 16ch). On the left is the distribution of normal force and on the right is the distribution of frictional force. (**a**) High friction condition, (**b**) low friction condition.

**Figure 8 sensors-24-04896-f008:**
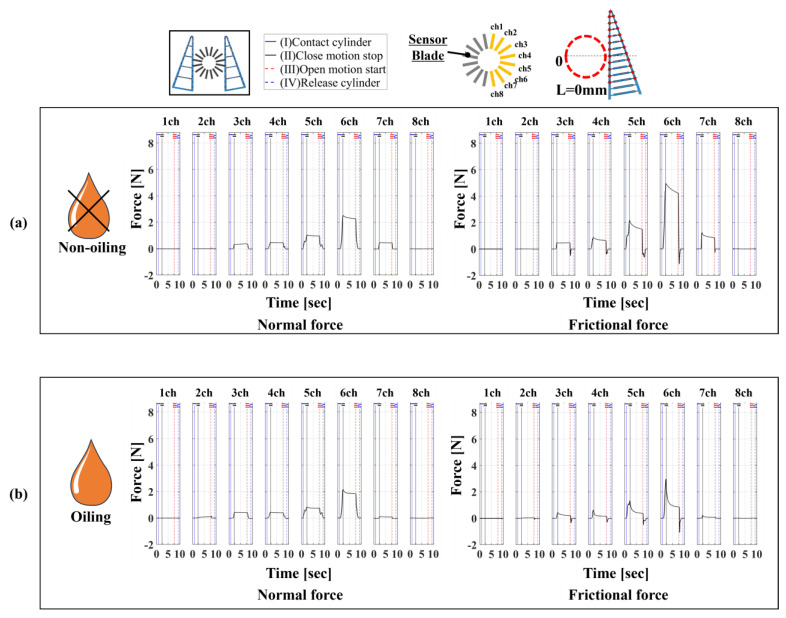
The 16/16-type measurement device using a 6-axis force sensor and the measurement method from 1ch to 8ch directions of 16 divisions (same from 9ch to 16ch). On the left is the distribution of normal force and on the right is the distribution of frictional force. (**a**) High friction condition, (**b**) low friction condition.

**Figure 9 sensors-24-04896-f009:**
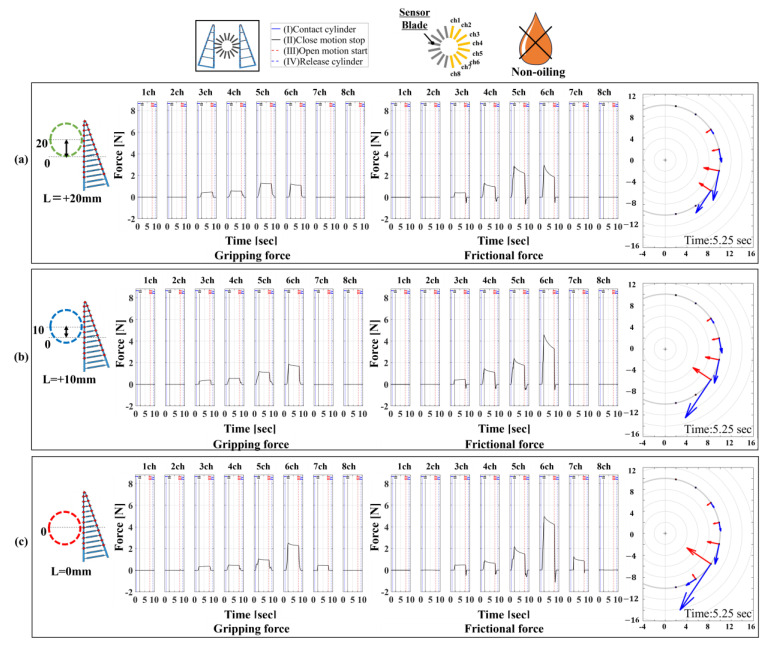
Normal and frictional force distribution and vector display for a high friction contact surface and cylindrical object under “fixed” conditions. (**a**) Contact start position: L = +20 mm, green; (**b**) contact start position: L = +10 mm, blue; and (**c**) contact start position: L = 0 mm, red.

**Figure 10 sensors-24-04896-f010:**
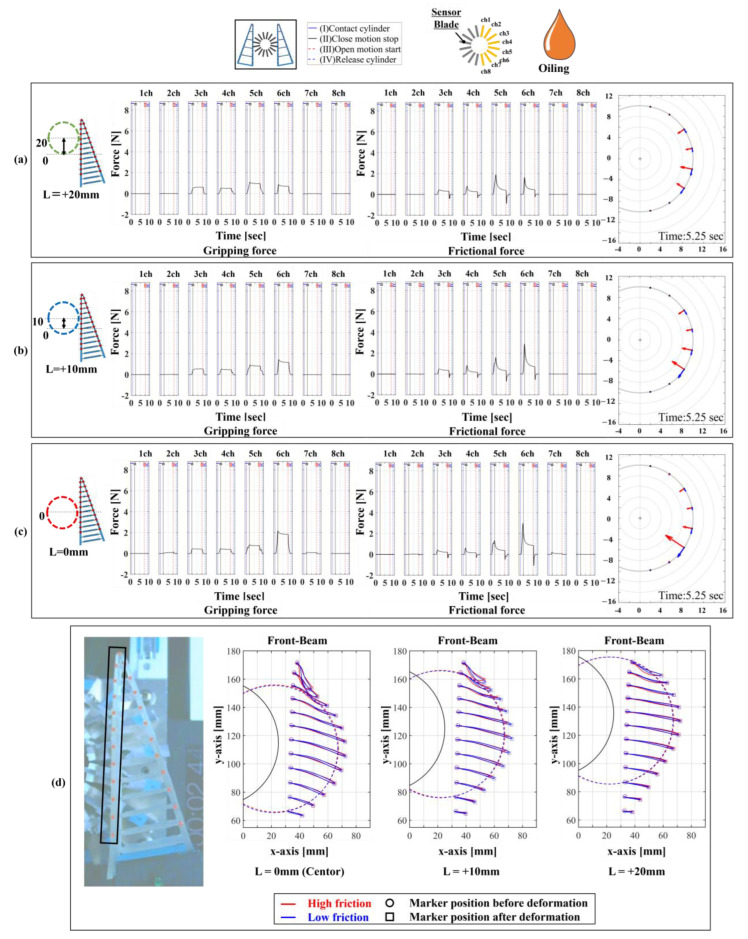
Normal and frictional force distribution and vector display for a low friction contact surface and cylindrical object under “fixed” conditions. (**a**) Contact start position: L = +20 mm, green; (**b**) contact start position: L = +10 mm, blue; (**c**) contact start position: L = 0 mm, red; and (**d**) structural deformation by marker tracking (high friction condition: red; low friction condition: blue).

**Figure 11 sensors-24-04896-f011:**
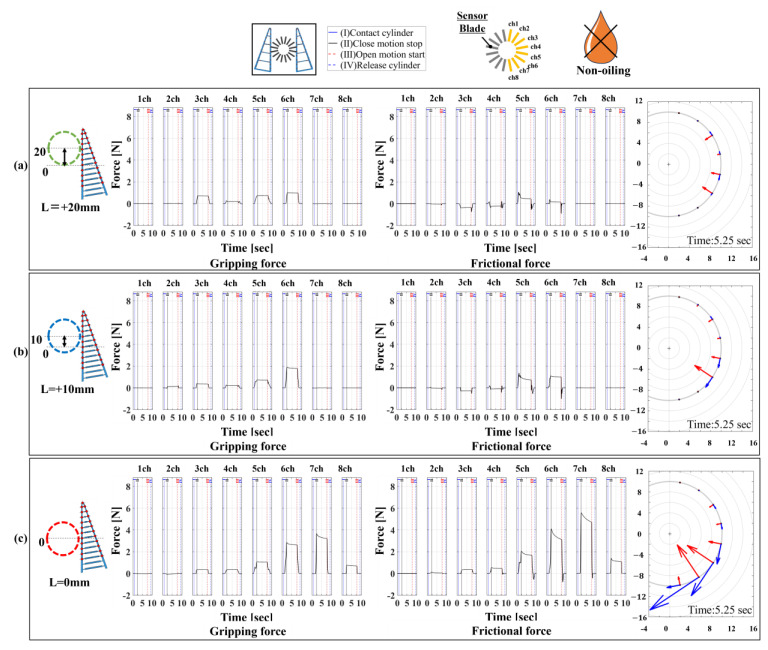
Normal and frictional force distribution and vector display for a high friction contact surface and cylindrical object under “non-fixed” conditions. (**a**) Contact start position: L = +20 mm, green; (**b**) contact start position: L = +10, blue mm; and (**c**) contact start position: L = 0 mm, red.

**Figure 12 sensors-24-04896-f012:**
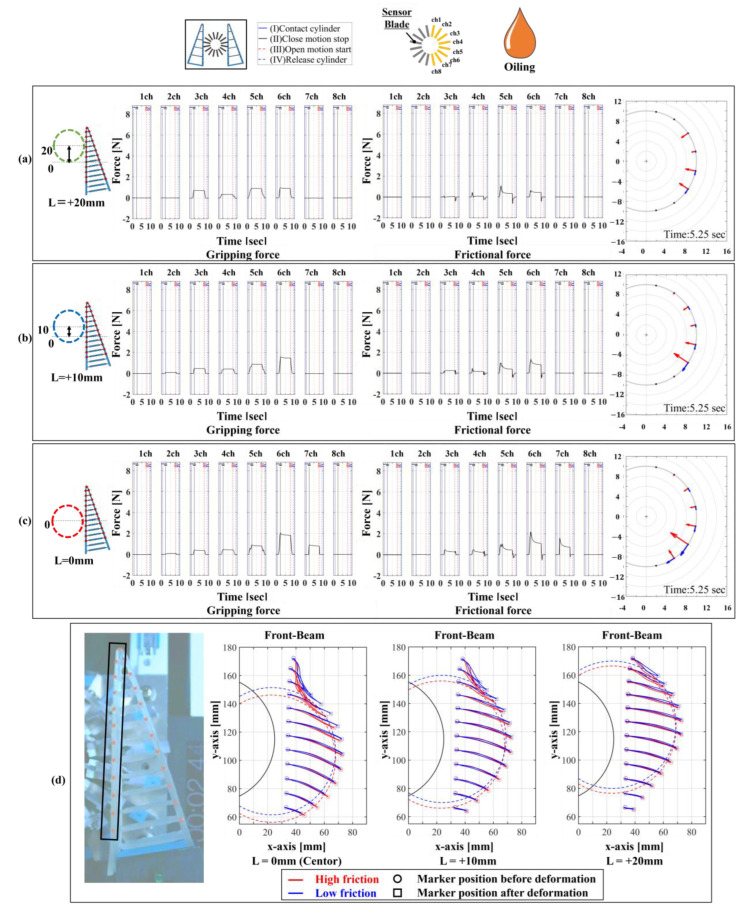
Normal and frictional force distribution and vector display for a low friction contact surface and cylindrical object under “non-fixed” conditions. (**a**) Contact start position: L = +20 mm, green; (**b**) contact start position: L = +10 mm, blue; (**c**) contact start position: L = 0 mm, red; and (**d**) structural deformation by marker tracking (high friction condition: red; low friction condition: blue).

**Figure 13 sensors-24-04896-f013:**
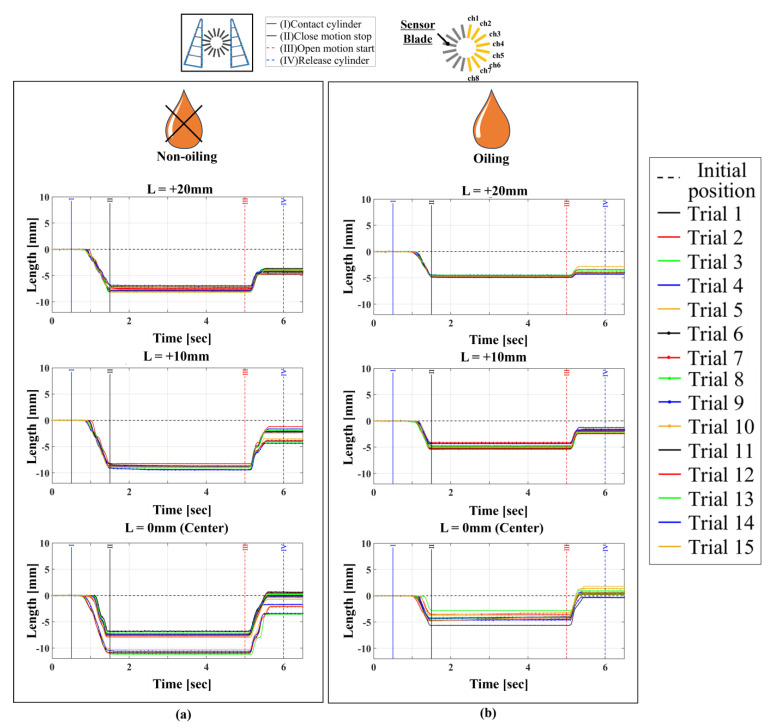
Displacement of the laser range finder in a movable object state under object-movable condition: “non-fixed” (top: L = +20mm / middle: L = +10mm / bottom: L = 0mm). (**a**) High friction condition; (**b**) low friction condition.

**Table 1 sensors-24-04896-t001:** Experimental conditions for a comparison of 1/16- and 16/16-type devices.

Condition Label	Device Type	Device Fixation Method	Contact Position	Contact Surface Friction Status	Figure Number
Setting 1	1/16	Fixed	Center (0 mm)	High (non-oiling)	Figure 7a
Setting 2	1/16	Fixed	Center (0 mm)	Low (oiling)	Figure 7b
Setting 3	16/16	Fixed	Center (0 mm)	High (non-oiling)	Figure 8a
Setting 4	16/16	Fixed	Center (0 mm)	Low (oiling)	Figure 8b

**Table 2 sensors-24-04896-t002:** Experimental setup with different gripping positions and contact surface conditions.

Condition Label	Device Type	Object Mounting Status	Contact Position	Contact Surface Friction Status	Figure Number
Setting 5	16/16	Fixed	Center + 20 mm	High (non-oiling)	Figure 9a
Setting 6	16/16	Fixed	Center + 10 mm	High (non-oiling)	Figure 9b
Setting 7	16/16	Fixed	Center (0 mm)	High (non-oiling)	Figure 9c
Setting 8	16/16	Fixed	Center + 20 mm	Low (oiling)	Figure 10a
Setting 9	16/16	Fixed	Center + 10 mm	Low (oiling)	Figure 10b
Setting 10	16/16	Fixed	Center (0 mm)	Low (oiling)	Figure 10c

**Table 3 sensors-24-04896-t003:** Experimental setup under conditions where objects can be moved.

Condition Label	Device Type	Object Mounting Status	Contact Position	Contact Surface Friction Status	Figure Number
Setting 11	16/16	Non-fixed	Center + 20 mm	High (non-oiling)	Figure 11a
Setting 12	16/16	Non-fixed	Center + 10 mm	High (non-oiling)	Figure 11b
Setting 13	16/16	Non-fixed	Center (0 mm)	High (non-oiling)	Figure 11c
Setting 14	16/16	Non-fixed	Center + 20 mm	Low (oiling)	Figure 12a
Setting 15	16/16	Non-fixed	Center + 10 mm	Low (oiling)	Figure 12b
Setting 16	16/16	Non-fixed	Center (0 mm)	Low (oiling)	Figure 12c

## Data Availability

The data presented in this study are available upon request from the corresponding author. The data are not publicly available due to the protection of intellectual property.

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
