# Peer review of "Development and Investigation of a Grasping Analysis System with Two-Axis Force Sensors at Each of the 16 Points on the Object Surface for a Hardware-Based FinRay-Type Soft Gripper"

_sensors, 2024, doi:10.3390/s24154896_

Round 1

Reviewer 1 Report

Comments and Suggestions for Authors

The introduction describes the workflow that should be included in the Materials and methods.

There is a lack of a clearly defined aim of the article.

Line 7 - You mention the word affiliation - it is not necessary to mention it.

Line 165 - From a statistical point of view, 5 trials is not a sufficient number for data interpretation.

ad. 1 In the comments, I state that the goal is not clearly defined.

ad. 2 The approach to solving the problem is original, i.e. design of the robotic head and its verification.

ad. 5 Since the goal of the work was not clearly defined, it could not be answered in the conclusion. At the end are collapsed heads - commonly available and newly designed. advertisement.

ad. 6 In my opinion, the links are appropriate

Reviewer 2 Report

Comments and Suggestions for Authors

This manuscript developed a 16-way object contact force measurement device incorporating 2-axis force sensors into each of the 16 segmented objects and compared the normal and tangential components of the enveloping grasping force of the FinRay soft gripper under two types of contact friction conditions. The proposed method focusing on the relationship between these forces in soft grippers. Therefore, some revisions should be adequately addressed before considering for publication.

1. In “Introduction” part, There are too many and too redundant conceptual introductions. It is suggested to streamline.

2. Figure 7a and 7b show that the maximum normal force in the high-friction and low-friction states was 2N in the 6ch direction and the frictional force decreased from about 5N to around 2N. Why such a result occurs, the author needs to discuss its reasons in depth.

3. How many experiments did the author conduct to obtain the corresponding data? If the sample size is relatively small, it cannot explain the problem. It is suggested to clarify the number of repeated experiments.

4. Why is excessive load generation reduced by conducting measurement in an experimental setting that considers non-fixed, back-and-forth movement of the gripped object, rather than fixed?

5. Figure 11, there are subfigures, the author should add the description in the article and caption of this figure. There is the same problem in figure 12.

6. The surface friction of soft gripper has a great influence on the acquisition of the experimental results. How does the author ensure the consistency of each measurement? 

7.The conclusion is too redundant. It is suggested to be streamlined.

Round 2

Reviewer 2 Report

Comments and Suggestions for Authors

The authors have addressed my issues.